# Morphology and Position of the Right Atrioventricular Valve in Relation to Right Atrial Structures

**DOI:** 10.3390/diagnostics11060960

**Published:** 2021-05-26

**Authors:** Jakub Hołda, Katarzyna Słodowska, Karolina Malinowska, Marcin Strona, Małgorzata Mazur, Katarzyna A. Jasińska, Aleksandra Matuszyk, Mateusz Koziej, Jerzy A. Walocha, Mateusz K. Hołda

**Affiliations:** 1HEART—Heart Embryology and Anatomy Research Team, Department of Anatomy, Jagiellonian University Medical College, 31-007 Cracow, Poland; jakub.p.holda@gmail.com (J.H.); kslodow11@gmail.com (K.S.); karolamali@wp.pl (K.M.); mmazur@uj.edu.pl (M.M.); kjasinska@uj.edu.pl (K.A.J.); amatuszyk@uj.edu.pl (A.M.); mateusz.koziej@gmail.com (M.K.); jwalocha@poczta.onet.pl (J.A.W.); 2Department of Forensic Medicine, Jagiellonian University Medical College, 31-007 Cracow, Poland; mstrona@uj.edu.pl; 3Department of Cardiovascular Sciences, University of Manchester, Manchester M13 9PL, UK

**Keywords:** coronary sinus ostium, right atrial appendage vestibule, terminal crest, tricuspid valve

## Abstract

The right atrioventricular valve (RAV) is an important anatomical structure that prevents blood backflow from the right ventricle to the right atrium. The complex anatomy of the RAV has lowered the success rate of surgical and transcatheter procedures performed within the area. The aim of this study was to describe the morphology of the RAV and determine its spatial position in relation to selected structures of the right atrium. We examined 200 randomly selected human adult hearts. All leaflets and commissures were identified and measured. The position of the RAV was defined. Notably, 3-leaflet configurations were present in 67.0% of cases, whereas 4-leaflet configurations were present in 33.0%. Septal and mural leaflets were both significantly shorter and higher in 4-leaflet than in 3-leaflet RAVs. Significant domination of the muro-septal commissure in 3-leflet valves was noted. The supero-septal commissure was the most stable point within RAV circumference. In 3-leaflet valves, the muro-septal commissure was placed within the cavo-tricuspid isthmus area in 52.2% of cases, followed by the right atrial appendage vestibule region (20.9%). In 4-leaflet RAVs, the infero-septal commissure was located predominantly in the cavo-tricuspid isthmus area and infero-mural commissure was always located within the right atrial appendage vestibule region. The RAV is a highly variable structure. The supero-septal part of the RAV is the least variable component, whereas the infero-mural is the most variable. The number of detected RAV leaflets significantly influences the relative position of individual valve components in relation to right atrial structures.

## 1. Introduction

The right atrioventricular valve (RAV) is an important anatomical structure of the human heart that prevents blood backflow from the right ventricle to the right atrium. RAV regurgitation is common, although it is mostly asymptomatic. It is typically diagnosed incidentally or when the pathology has led to severe clinical symptoms [1]. Disorders of the RAV are mainly treated with surgical techniques [2]. However, these procedures are challenging and have a high risk of complications [2,3,4]. Therefore, clinicians are exploring less invasive treatment alternatives [5,6]. Several promising techniques and devices are currently in development. These include the FORMA spacer device, the Trialign device, the Caval Valve Implantation (CAVI), the TriClip device and many others [5,7,8,9]. It is anticipated that minimally invasive procedures will become routine options for patients with contraindications to classical cardiac surgery [6,7]. 

The RAV is a sophisticated morphological entity. The complex anatomy of the RAV has lowered the success rate of surgical and transcatheter procedures performed within the area [10,11]. For instance, although the RAV is commonly referred to as the tricuspid valve, a recent functional anatomy study found that almost 40% of examined specimens had four distinct leaflets [10]. The infero-mural region of the valve is the most variable area of right atrioventricular valve. The RAV had a lot more diversity in the shapes and sizes of its scallops, commissures, tendinous chords and papillary muscles compared to mitral valve [10,12]. The RAV leaflets were found to be anatomically divided into scallops in the irregular fashion with the septal leaflet being the most subdivided part of the RAV. Additionally, the functional scallops may be found within the RAV leaflets (mainly in septal and mural leaflets) [10]. Furthermore, the RAV annulus was found to have an irregular and nonplanar shape [10,13]. Nevertheless, no significant differences in the overall annular sizes between valves eliciting different leaflet configurations were observed in the previous study [10]. Finally, there was even significant variability in the anatomic landmarks and locations of each leaflet and commissure [10]. Understanding the detailed morphology of the RAV and the spatial relationships between important anatomic landmarks is crucial for the success of invasive cardiac procedures. The RAV annulus size and geometry has a crucial impact on surgical repair procedures, while leaflets and commissure configurations, together with their location within the RAV orifice, influence transcatheter procedures, especially with coaptation devices [5,9].

Therefore, the aim of this study was to gain a better understanding of the anatomy of the RAV and describe the spatial relationships between selected structures of the right atrium. We hope that the collected data will help to enhance the quality and safety of cardiac surgeries and the minimally invasive transcatheter procedures conducted within the RAV area.

## 2. Methods

This study was approved on 23 April 2020 by the Bioethical Committee of the Jagiellonian University in Cracow, Poland (No. 1072.6120.90.2020), and its study protocol conforms to the ethical guidelines of the 1975 Declaration of Helsinki.

### 2.1. Study Population

We examined 200 random human adult hearts. The specimens were collected during routine forensic autopsies. The hearts were taken from Caucasian individuals (of which 22.0% were female) ranging in age from 18 to 94 (mean age was 46.9 ± 17.9 years). The donors had a mean body mass index (BMI) of 26.6 ± 4.5 kg/m^2^ and an average body surface area (BSA) of 1.9 ± 0.2 m². Their primary causes of death were suicide, murder, traffic accidents and home accidents. We excluded donors with severe anatomical defects, heart trauma, heart grafts, severe macroscopic pathologies and those with signs of cadaveric decomposition.

### 2.2. Dissection and Measurements

The hearts and the proximal parts of the main vessels were dissected from each chest cavity in a routine manner. Before being immersed in 10% paraformaldehyde solution, the hearts were weighed using a 0.5 g precision electronic laboratory scale (SATIS, BSA-L Laboratory, Poland). The right atrium was opened through an intercaval incision extending from the orifice of the superior vena cava to the orifice of the inferior vena cava.

The detailed morphology of each valve was examined. We identified, labelled, and counted all the commissures and leaflets. A leaflet was defined as a tissue fold located between two commissures. Using the attitudinally correct terminology of human anatomy, we identified 4 leaflets: a superior, a septal, a mural and an inferior leaflet (although the latter was only present in hearts with a 4-leaflet configuration) [10,14]. Commissures were defined as areas within the valve located between two adjacent leaflets. They had a characteristic arched appearance and were accompanied by a fan-shaped cord originating from the respective papillary muscle (or septal band) located directly below the commissure [10,15,16]. We observed the following commissures: the supero-septal commissure (found between the superior and the septal leaflet); the supero-mural commissure (found between the superior and the mural leaflet); the muro-septal commissure (found between the mural and the septal leaflet in the 3-leaflet configuration); the infero-septal commissure (found between the inferior and the septal leaflet in the 4-leaflet configuration) and the infero-mural commissure (found between the inferior and the mural leaflet in the 4-leaflet configuration) (Figure 1) [10].

We measured the length of each leaflet and commissure bordering the hinge line of the RAV annulus (Figure 1). The perimeter of the annulus was calculated as the sum of the lengths of the leaflets and commissures. Leaflet height was defined as the maximum distance from the hinge line to the free edge (see Figure 1). To determine the spatial position of the RAV, we described the location of the septal leaflet and its adjacent commissures with respect to selected landmarks within the right atrium. We measured the distance from the terminal crest (the inferior isthmus of the right atrial appendage vestibule) to the muro-septal commissure (in the 3-leaflet configuration) and from the terminal crest to the infero-septal commissure in the 4-leaflet configuration (see A). We also measured the distance from the center point of the ostium of the coronary sinus to the center point of the supero-septal and the muro-septal commissures (in hearts with a 3-leaflet configuration) or the infero-septal commissure (in hearts with a 4-leaflet configuration) (see B and C). Lastly, we measured the distance from the superior isthmus of the right atrial appendage vestibule to the supero-septal commissure and to the supero-mural commissure (see D and E) (Figure 1). 

Linear measurements were obtained using 0.03 mm precision electronic calipers (YATO, YT–7201, Wrocław, Poland). All measurements were performed by two independent researchers to reduce human bias. If results obtained by the two researchers differed by more than 10%, the sample was reassessed. The mean of the two new measurements was calculated, approximated to the tenth decimal place, and reported as the new final value.

### 2.3. Statistical Analysis

The data are presented as the mean ± the standard deviation (SD), along with the minimum and maximum values (range) for continuous variables, or as percentages (%) for categorical variables. A *p*-value < 0.05 was considered statistically significant, Shapiro–Wilk tests were used to determine if the quantitative data were normally distributed. Levene’s test was performed to verify relative homogeneity of variance. Student’s *t*-tests and the Mann–Whitney U tests were used for statistical comparisons. Correlation coefficients were calculated; to detect a simple correlation (*r* = 0.25) with 80% power and a 5% significance level (two-tailed; α = 0.05; β = 0.2), the minimal sample size was set at 123 cases. The analysis of variance (ANOVA) or non-parametric Kruskal–Wallis test was used to compare values between different groups. Detailed comparisons were performed using Tukey’s post hoc analyses. Qualitative variables were compared using χ2 (chi-squared) tests of proportions with Bonferroni corrections to account for the multiple comparisons. Statistical analyses were performed using StatSoft STATISTICA 13.1 software (StatSoft Inc., Tulsa, OK, USA).

## 3. Results

### 3.1. The RAV Morphology

Results are presented in Table 1. In 67.0% of specimens, the RAV had a 3-leaflet configuration (Figure 2A). In the remaining 33.0% of hearts, the RAV had four leaflets (Figure 2B). Valves with four leaflets had significantly larger perimeters than those with three leaflets (119.2 ± 11.1 vs. 109.3 ± 11.3 mm, *p* = 0.001). No difference in heart weight was observed between two configurations (407.9 ± 86.5 vs. 419.3 ± 75.8 g, *p* = 0.363). There was a correlation between the perimeter of the valve and the donors’ BMI in 3-leaflet heart valves (*r* = 0.31, *p* < 0.001), but no such association was present in the 4-leaflet configuration (*r* = 0.01, *p* = 0.922). In hearts with a 3-leaflet configuration, the lengths of the superior leaflet and of the septal leaflet were similar, whereas the length of the mural leaflet was about twice as short (Table 1, *p* > 0.05). The same trends were noted in 4-leaflet heart valves (Table 1, *p* > 0.05). Interestingly, the individual leaflet height did not differ much within the same valve configuration (Table 1, *p* > 0.05). When present, the inferior leaflet was comparable in size to the mural leaflet (both in length and in height) (Table 1, *p* > 0.05). Significant differences in leaflet size were observed between the two valve configurations (Table 1). The superior leaflet had the least variability, and its length and height were comparable in 3- and 4-leaflet valves. Meanwhile, the septal and mural leaflets were both significantly shorter and higher in the 4-leaflet RAVs (Table 1).

Commissure lengths varied considerably within the 3-leaflet heart valves. The muro-septal commissure was substantially larger than the two remaining commissures (Table 1, *p* < 0.001). Such length disparities were not observed in the 4-leaflet configuration, where all four commissures had comparable lengths (Table 1, *p* > 0.05). Furthermore, the supero-mural commissure was significantly longer in the 3-leaflet heart valve when compared to the 4-leaflet heart valve (9.9 ± 4.0 vs. 7.5 ± 2.2 mm, *p* < 0.001). 

Age, sex and heart weight variables did not influence RAV configuration, leaflet size or commissure length (all *r* < 0.2 and *p* > 0.05). 

### 3.2. The Spatial Position of the RAV

The number of detected leaflets in the RAV was found to affect the relative position of the valve components. The supero-septal commissure was located on the septal side of the right atrium in proximity to the apex of Koch’s triangle, and its location was the most stable point within the RAV circumference. Its position remained unchanged when comparing the 3- and 4-leaflet heart valves (Table 1, *p* > 0.05). In all examined hearts, the supero-septal commissure was located on the left-superior side of the ostium of the coronary sinus (Figure 1). The supero-septal commissure was never present within the vestibule of the right atrial appendage. This contrasted with the location of the supero-mural commissure, which was always found in the vestibule’s middle sector. We observed a strong positive correlation between the length of the supero-septal commissure and the distance between the commissure and the ostium of the coronary sinus (*r* = 0.51, *p* < 0.001). No association was detected between the distance from the commissure to the ostium of the coronary sinus and the septal leaflet length (*r* = 0.12, *p* = 0.31).

The location of commissures found within the infero-mural aspect of the RAV annulus (the muro-septal and the infero-septal commissures) varied considerably, not only between different RAV configurations but also between individual hearts with the same number of RAV leaflets. In the 3-leaflet heart valves, the muro-septal commissure was located further away from the ostium of the coronary sinus than the infero-septal commissure in the 4-leaflet configuration (16.1 ± 9.0 vs. 7.7 ± 6.3 mm, *p* < 0.001). The opposite trend was observed for the commissure’s distance to the terminal crest (5.8 ± 9.4 vs. 13.5 ± 6.4 mm, respectively; Table 1, *p* < 0.001). The location of the muro-septal commissure differed significantly: in 52.2% of specimens, it was located in the area of the cavo-tricuspid isthmus (situated between the ostium of the coronary sinus and the terminal crest); in 9.0% of hearts, it was situated at the level or to the left of the ostium of the coronary sinus; in 17.9%, it was at the level of the terminal crest; in the remaining 20.9% of cases, it was found within the vestibule of the right atrial appendage (Figure 3). Additionally, there was a positive correlation between the segment from the muro-septal commissure to the terminal crest and the superior leaflet length (*r* = 0.32, *p* = 0.008). Meanwhile, this same segment had a negative correlation with septal leaflet length (*r* = −0.38, *p* = 0.001).

In 4-leaflet RAVs, the infero-septal commissure was predominantly located in the cavo-tricuspid isthmus area (81.8% of cases). It was never located within the area of the vestibule of the right atrial appendage. In the remaining 12.1% of hearts, the commissure was located at the level of the ostium of the coronary sinus, while in 6.1% of cases, it pointed towards the septal direction. The spatial position of the infero-septal commissure in the 4-leaflet valve was influenced by the length of the septal leaflet (there was a positive correlation with the commissure’s distance to the ostium of the coronary sinus (*r* = 0.51, *p* = 0.03) and a negative correlation with the commissure’s distance to the terminal crest (*r* = −0.37, *p* = 0.03). The infero-mural commissure was always located within the area of the right atrial appendage vestibule.

The relative positions of the components of the RAV were not affected by the donors’ age, sex, cardiac weight or any other anthropometric features (weight, height, BMI) (all *r* < 0.2 and *p* > 0.05).

## 4. Discussion

Despite its important function, the RAV is often considered as the “forgotten valve”. However, our study further shows the complexity and variability of the RAV’s morphology. This valve is much more irregular than its sister valve, the mitral valve, which has a remarkably more simple and predictable structure [15]. While the mitral valve is almost always composed of a 2-leaflet configuration, the RAV often presents with a 4-leaflet configuration. In the scientific literature, the RAV is often referred to as the “tricuspid valve”, although we strongly suggest avoiding this nomenclature, since it may mistakenly suggest the valve’s construction [10]. Moreover, before attempting any interventions within the RAV region, it is important to determine the amount of leaflets using either echocardiography or cardiac computed tomography [17,18,19]. Finally, it is important to know that there are considerable differences in leaflet and commissure sizes between 3- and 4-leaflet RAVs (Table 1). Our study has shown that the supero/septal region of the RAV is the least variable area, whereas the infero/mural region differs significantly between 3-and 4-leaflet RAVs. It seems that the formation of the additional fourth leaflet takes place at the expense of the size of the muro-septal commissure and of the septal leaflet, which also explains the shifted position of the muro-septal and infero-septal commissures (Table 1). 

Besides the variable morphology of the RAV’s leaflets and commissures, there are other causes that affect the valve’s hemodynamic characteristics. Taken together, they make RAV imaging and interventions (both surgical and percutaneous) challenging [13]. The RAV annulus is a relatively large, flexible, fragile and highly irregular structure, and it is easily affected by changes in the shape and size of the right ventricle [20]. The nonplanarity of the RAV annulus and its heterogeneous regional behavior may have a strong association with leaflet configuration. This study has shown that there is a significant difference in the size of the annulus between 3- and 4-leaflet valves. Initially larger right atrioventricular orifice size may force the creation of the additional leaflet and commissure to seal the orifice that are located with the inferior/mural region of the annulus [10,19]. 

When planning therapeutic procedures on the RAV, it is important to determine its morphology, position and neighboring cardiac structures [17]. The RAV annulus has important anatomical relationships with the right coronary artery, the acute marginal branch, the small cardiac vein (located within the vestibule of the right atrial appendage), the terminal crest, the cavo-tricuspid isthmus, the coronary sinus (and its ostium), the atrio-ventricular node and the aortic valve [21,22,23,24]. The above-mentioned structures are subject to injury, and clinicians must factor in their respective positions before attempting any invasive procedures. Although the inferior/mural region of the RAV has the most unpredictable morphology, it should be considered the safest area for interventions due to its remoteness from crucial heart structures (the only exception being the right coronary artery) [25]. 

Transcatheter RAV annulus reduction techniques involve placing special pledgeted sutures on the superior and mural leaflet. These procedures reduce the entire circumference of the valve annulus and seal the orifice. This, in turn, improves the RAV’s hemodynamic properties [26,27]. Our observations have shown that the location of the supero-mural commissure is quite consistent. It is usually located near the middle isthmus of the vestibule of the right atrial appendage and is relatively easy to discern. However, the locations of the commissures and leaflets of the mural/inferior region of the valve are not always clear. Even RAVs with the same number of leaflets present with a lot of heterogeneity (see Table 1). In 3-leaflet valves, the larger septal leaflet ends much closer to the terminal crest and further away from the ostium of the coronary sinus when compared to 4-leaflet RAVs. The muro-septal commissure in a 3-leaflet valve may be located anywhere between the left side of Koch’s triangle and the right side of the middle isthmus of the right atrial appendage vestibule. In 4-leaflet valves, the smaller septal leaflet pushes the inferior leaflet to the left into the territory of the cavo-tricuspid isthmus but never to the region of the vestibule of the right atrial appendage. The presence of an additional inferior leaflet and its adjacent commissures shifts the mural leaflet to the right side of the valve. Therefore, we recommend determining the detailed morphology of leaflet valves before qualifying patients for invasive procedures. It is equally important to study the course of coronary vessels within the RAV region in order to avoid any damage to these vessels [17,25,28].

There are some limitations to this study. The main one is that this study was performed on autopsied material preserved in paraformaldehyde solution. Consequently, there may have been slight changes to the size and shape of the studied hearts due to fixation. Nevertheless, our earlier studies have demonstrated that using paraformaldehyde solution did not cause important changes in cardiac dimensions [29,30]. Furthermore, since we analyzed post-mortem material, we were not able to assess the behavior and dimensional changes of RAV components during the cardiac cycle. Additionally, we only investigated hearts from healthy donors and, therefore, our results may not be applicable to patients with structural valvular diseases. Despite these limitations, we consider that this study has provided insight into the morphological and topographical analyses of the RAV, as well as into the relationships between individual RAV components and right atrial structures.

## Figures and Tables

**Figure 1 diagnostics-11-00960-f001:**
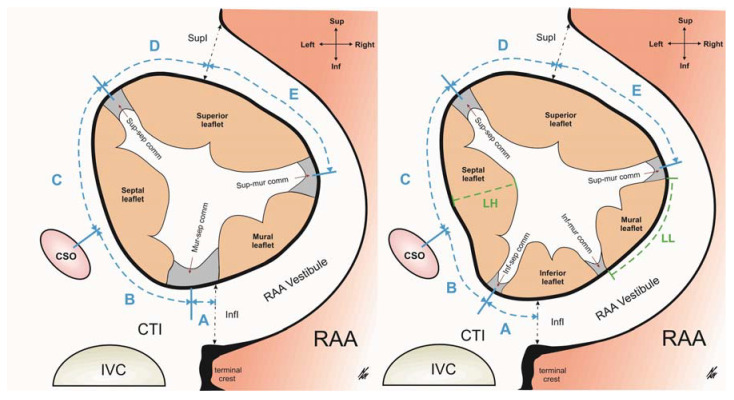
Schematic views of the RAV in 3-leaflet and 4-leaflet configurations. Performed measurements are marked (A—terminal crest to muro-septal or infero-septal commissure, B—coronary sinus ostium to supero-septal commissure, C—coronary sinus ostium to muro-septa or infero-septal commissure, D—right atrial appendage vestibule superior isthmus to the supero-septal commissure, and E—right atrial appendage vestibule superior isthmus to supero-mural commissure) with the relative proportions preserved. CSO—coronary sinus ostium, CTI—cavo-tricuspid isthmus, InfI—inferior isthmus, IVC—inferior vena cava, LH—leaflet height, LL—leaflet length, RAA—right atrial appendage, SupI—superior isthmus.

**Figure 2 diagnostics-11-00960-f002:**
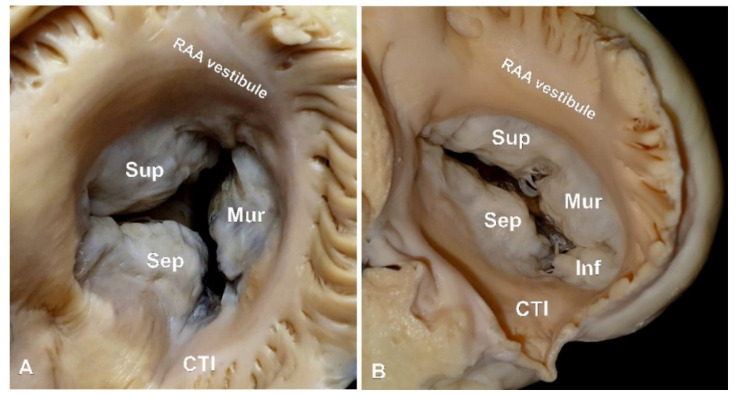
Photographs of a cadaveric heart specimens showing examples of (**A**) 3-leaflet and (**B**) 4-leaflet right atrioventricular valve. CTI—cavo-trisuspid isthmus, RAA—right atrial appendage.

**Figure 3 diagnostics-11-00960-f003:**
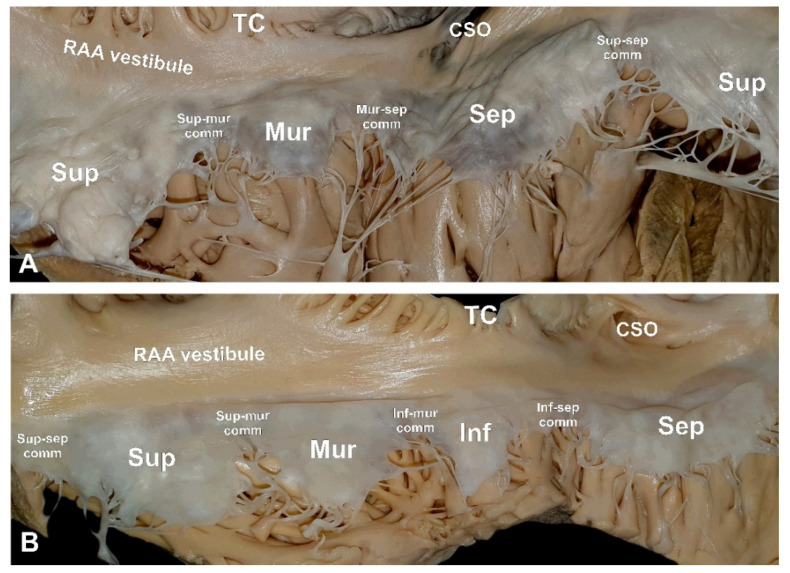
Photographs of cadaveric heart specimens showing examples of (**A**) 3-leaflet and (**B**) 4-leaflet right atrioventricular valves. CSO—coronary sinus ostium, RAA—right atrial appendage, TC—terminal crest.

**Table 1 diagnostics-11-00960-t001:** Morphometric observations of structure and position of right atrioventricular valve in 3-leaflet and 4-leaflet configurations (mean ± SD with range).

Parameter		3-Leaflet Valves (*n* = 134)	4-Leaflet Valves (*n* = 66)	*p*-Value
Sex (% females)		20.9	24.2	0.597
Age (years)		47.1 ± 18.2 (18–94)	46.5 ± 18.4 (21–87)	0.827
BMI (kg/m^2^)		26.3 ± 4.8 (15.2–39.6)	27.5 ± 4.1 (17.4–37.8)	0.083
Heart weight (g)		407.9 ± 86.5 (230.0–606.5)	419.3 ± 75.8 (274.5–620.0)	0.363
Valve perimeter (mm)		109.3 ± 11.3 (86.6–156.8)	119.2 ± 11.1 (96.5–143.4)	**<0.001**
Leaflet length (mm)	superior leaflet	30.1 ± 9.2 (12.6–71.0)	28.8 ± 9.0 (11.5–50.5)	0.09
	septal leaflet	30.5 ± 5.7 (17.7–46.6)	27.0 ± 5.7 (17.8–39.4)	**0.005**
	mural leaflet	17.5 ± 6.2 (5.9–44.5)	14.8 ± 5.5 (7.2–28.0)	**<0.001**
	inferior leaflet *	-	14.3 ± 4.6 (8.2–26.5)	-
Leaflet height (mm)	superior leaflet	12.6 ± 4.2 (3.5–23.7)	13.7 ± 5.1 (4.6–26.7)	0.11
	septal leaflet	11.8 ± 3.9 (3.3–20.8)	14.3 ± 4.5 (5.2–22.8)	**<0.001**
	mural leaflet	11.9 ± 3.6 (4.0–20.7)	13.6 ± 4.0 (4.4–23.8)	**0.003**
	inferior leaflet *	-	12.1 ± 8.6 (2.3–19.6)	-
Commissure length (mm)	supero-septal	9.6 ± 3.4 (5.1–21.5)	8.6 ± 4.7 (4.2–24.5)	0.17
	supero-mural	9.9 ± 4.0 (3.5–21.4)	7.5 ± 2.2 (4.0–14.3)	**<0.001**
	muro-septal **	14.0 ± 4.8 (4.5–25.9)	-	-
	infero-septal *	-	7.9 ± 3.0 (2.9–18.2)	-
	infero-mural *	-	7.7 ± 2.4 (3.5–13.5)	-
A—RAAV inferior isthmus to muro-septal ** or infero-septal * commissure (mm) ^†^		5.8 ± 9.4 (−15.6–29.7)	13.5 ± 6.4 (0.0–27.9)	**<0.001**
B—coronary sinus ostium to muro-septal ** or infero-septal * commissure (mm) ^‡^		16.1 ± 9.0 (−12.7–33.2)	7.7 ± 6.3 (-10.5–22.6)	**<0.001**
C—coronary sinus ostium to supero-septal commissure (mm)		20.1 ± 5.6 (5.3–36.7)	20.5 ± 5.1 (9.8–29.2)	0.62
D—RAAV superior isthmus to supero-septal commissure (mm)		7.3 ± 3.8 (1.3–14.3)	6.2 ± 5.3 (1.4–14.8)	0.09
E—RAAV superior isthmus to supero-mural commissure (mm)		33.9 ± 11.2 (17.5–72.7)	31.1 ± 9.6 (11.9–51.6)	0.08

*—n 4-leaflet configuration only; **—in 3-leaflet configuration; ^†^—negative values represent commissures located within the right atrial appendage vestibule; ^‡^—negative values represent commissures located closer to the septal region than coronary sinus ostium; Figure 1 shows schematic sites of measurements. RAAV—right atrial appendage vestibule. Bold values denote statistical significance.

## Data Availability

Data are available from the authors upon reasonable request.

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
