# Peer review of "Morphology and Position of the Right Atrioventricular Valve in Relation to Right Atrial Structures"

_diagnostics, 2021, doi:10.3390/diagnostics11060960_

Round 1

Reviewer 1 Report

In this study the authors provide a modified nomenclature to label the right atrioventricular valve leaflet description , they already published  (ref 10) in a larger cohort of heart specimen. The study is interesting well described. Somme comments have to be addressed.

Could you explain why not define the 4th leaflet  as a cleft of the inferior leaflet? by the way whatever  the findings are interesting

If I’m not mistaken in the previous published paper the authors referred the 4° leaflet as “functional”. Could the authors develop

What is the hypothesis of the authors about the perimeters of the valve that was not different in their previous serie (No difference in valve perimeters between 2 valve types were observed (112.2 vs. 117.1 mm; p = 0.14)  whereras in the present study  Valves with four leaflets had significantly larger perimeters than those with three leaflets (119.2 135 ± 11.1 vs. 109.3 ± 11.3 mm, P = 0.001).

Could the authors described briefly  in the figure legend the measurements A BC..

Although Age and sex variables did not influence RAV configuration,

Could the authors report them in the table the age, sex and BMI

In the discussion section   the authors reported “this valve is much more irregular than its sister valve, the bicuspid valve”,, the authors mean the Mitral  valve

Author Response

Reviewer: 1 

In this study the authors provide a modified nomenclature to label the right atrioventricular valve leaflet description , they already published  (ref 10) in a larger cohort of heart specimen. The study is interesting well described. Somme comments have to be addressed.

Dear Reviewer, Thank you very much for your revision and comments. We are glad to hear that our article was interesting for You. Thank you also for noticing its innovative nature. We hope that our anatomical observations will be the basis for further clinical studies and will help perform procedures within this region. Please find below our responses to your specific comments. Yours sincerely Authors  

Could you explain why not define the 4th leaflet  as a cleft of the inferior leaflet? by the way whatever  the findings are interesting

  • There is a considerable difference between this two structures. The term cleft may be used only for congenital, pathologic leaflet splits that may be located in any leaflet and (1) reach the valve annulus or (2) are not connected with the tendinous cords to the papillary muscle. The observed 4th leaflet have its own supporting commissure and commissural tendinous cords as other leaflets within the valve.

If I’m not mistaken in the previous published paper the authors referred the 4° leaflet as “functional”. Could the authors develop

  • All identified in the previous study valve leaflets were functional. Previously, we have defined a leaflet as a fold of tissue that was located between 2 commissures. These were independent functional components of a given valve that reached the geometrical center of the valve, coapted along the zones of apposition, and touched at least 2 other leaflets during systole. Because in the previous work the “functioning” (in the beating heart) right atrioventricular valves were analyzed using endoscopic cameras we were able to describe not only the anatomy, but also a function of the valve. In the current study only the post mortem static examination was possible, thus we abandoned to use the term “functional”. Nevertheless, based on our previous study we are sure that all leaflets (both in 3- and 4-leaflets configuration) are functional leaflets according to the above mentioned definition.

What is the hypothesis of the authors about the perimeters of the valve that was not different in their previous serie (No difference in valve perimeters between 2 valve types were observed (112.2 vs. 117.1 mm; p = 0.14)  whereras in the present study  Valves with four leaflets had significantly larger perimeters than those with three leaflets (119.2 135 ± 11.1 vs. 109.3 ± 11.3 mm, P = 0.001).

  • This is most likely due to 5 times larger study population included into the current study, thus the variability of included donors and the statistical power of the observations increased significantly.

Could the authors described briefly in the figure legend the measurements ABC...

  • This is now described in Figure 1 legend.

Although Age and sex variables did not influence RAV configuration, could the authors report them in the table the age, sex and BMI

  • This is now displayed in table 1

In the discussion section   the authors reported “this valve is much more irregular than its sister valve, the bicuspid valve”,, the authors mean the Mitral  valve

  • Yes, this is now corrected.

Reviewer 2 Report

The manuscript by HoÅ‚da et al is in effect a follow-up study of HoÅ‚da et al (2019, JACC: Cardiovascular Interventions) and gives a more detailed quantitative description of the right atrioventricular valve than, I believe, has been published to date. A substantial number – 200 – of human autopsy hearts have been investigated. It is concluded, in agreement with the previous study, that the tricuspid valve in fact is frequently configured as 4 leaflets. The novelty, is then the finding that the 4-leaflet configuration associates with a slightly larger orifice and greater dimensions to some of the leaflets. In addition, it is found the mural leaflets and associated structures are more variable in position than the septal leaflet, irrespective of the valve configuration is 3 or 4 leaflets. The text is concise and the figures are clear.

The presentation can arguably be improved by the use of additional images to firstly better document the differences between the 3- and 4 leaflet configurations and secondly to graphically summarize the Results. More importantly, it is not clear at this point whether the reported differences in dimensions of the 3-leaflet and 4-leaflet configurations can be explained by differences in heart size (rather than being intrinsic to the two configurations). In addition, we have little insight into how reliably the 3-leaflet and a 4-leaflet configurations can be distinguished.   

Both the Introduction and the Discussion attempt to contextualize to surgical interventions and while annulus size is of obvious importance, it does not become clear why a 3-leaflet or a 4-leaflet configuration would make a difference to the corrective surgery (as is implied). Please amend.

In effect, the present manuscript is a follow-up study on Hołda et al 2019 (JACC: Cardiovascular Interventions, 12(2), 169-178). It would then make sense if the Introduction briefly summarized what was learned in the previous study so as to better set the stage for what we are going to learn in the present manuscript.

The figures are clear and illustrative. Nonetheless, it would be very helpful if;

In Figure 1, if you leave out the oval fossa, you can fit in an additional schematic namely of a 3-leaflet valve (say on the left). Also, at present I find it difficult to understand whether the hinge line between A and E was measured and if it wasn’t, why not?

In Figure 2 to have a photo of a 3-leaflet valve juxtaposed to the current photo

In Figure 3 to have a photo of a 4-leaflet valve juxtaposed to the current photo.

Please also consider adding a photo of an ambiguous case; as you know, one person’s commissure (=2 leaflets) is another person’s cleft (=1 leaflet).

Please consider using a figure like Figure 1 (both 3-leaflet and 4-leaflet schematic), to report your findings (analogous to using a ‘bulls eye plot’ to report on LV 17-segments). Although your text in Results is concise, there is still a substantial number of values and anatomical locations to keep track of and it may be substantially easier if reported in schemes (bold or red font for significantly different values?). Please consider reporting median values (is that not more useful than mean?).

Do I understand correctly that you have heart weights for each heart but in the linear measurements (Table 1) you have not corrected for heart size? Why not? Are the 4-leaflet hearts larger and may that explain the great dimensions? You do correlate valve perimeter to BMI split on the 3- and 4-leaflet configuration, but why not first do the analysis on all hearts against all BMI, and heart weight? Also, how do your measurements of RAV perimeter (if corrected for body mass) compare to those reported previously (Rowlatt, U. F., Rimoldi, H. J., & Lev, M. (1963). The quantitative anatomy of the normal child's heart. Pediatric Clinics of North America, 10(2), 499-588.).

Even if you find no significant effect or correlation, please still report the output of the statistical tests, e.g. for correlation r and p. For example concerning 4-leaflet configuration (L138), age and sex (L164-165) – as you have done elsewhere (L179).

Are there in vivo data (echocardiography, MRI, CT) that support your morphometric findings?

L47-48: “more” – compared to what, the mitral valve?

L111: okay for linear measurements, but how did you measure curves (hinge lines)?

Please note that references 10 and 16 are the same. Please remove one and update the references.

Author Response

Reviewer: 2 

The manuscript by HoÅ‚da et al is in effect a follow-up study of HoÅ‚da et al (2019, JACC: Cardiovascular Interventions) and gives a more detailed quantitative description of the right atrioventricular valve than, I believe, has been published to date. A substantial number – 200 – of human autopsy hearts have been investigated. It is concluded, in agreement with the previous study, that the tricuspid valve in fact is frequently configured as 4 leaflets. The novelty, is then the finding that the 4-leaflet configuration associates with a slightly larger orifice and greater dimensions to some of the leaflets. In addition, it is found the mural leaflets and associated structures are more variable in position than the septal leaflet, irrespective of the valve configuration is 3 or 4 leaflets. The text is concise and the figures are clear.

The presentation can arguably be improved by the use of additional images to firstly better document the differences between the 3- and 4 leaflet configurations and secondly to graphically summarize the Results. More importantly, it is not clear at this point whether the reported differences in dimensions of the 3-leaflet and 4-leaflet configurations can be explained by differences in heart size (rather than being intrinsic to the two configurations). In addition, we have little insight into how reliably the 3-leaflet and a 4-leaflet configurations can be distinguished.   

  •  Dear Reviewer, Thank you very much for your revision and comments. We are glad to hear that our article was interesting for You. Thank you also for noticing its innovative nature. We hope that our anatomical observations will be the basis for further clinical studies and will help perform procedures within this region. Please find below our responses to your specific comments. Yours sincerely Authors

Both the Introduction and the Discussion attempt to contextualize to surgical interventions and while annulus size is of obvious importance, it does not become clear why a 3-leaflet or a 4-leaflet configuration would make a difference to the corrective surgery (as is implied). Please amend.

  • The introduction section was corrected and expanded to differentiate between surgical and transcatheter procedures.

In effect, the present manuscript is a follow-up study on Hołda et al 2019 (JACC: Cardiovascular Interventions, 12(2), 169-178). It would then make sense if the Introduction briefly summarized what was learned in the previous study so as to better set the stage for what we are going to learn in the present manuscript.

  • The introduction section was expanded presenting key results of our previous study.

The figures are clear and illustrative. Nonetheless, it would be very helpful if; In Figure 1, if you leave out the oval fossa, you can fit in an additional schematic namely of a 3-leaflet valve (say on the left). Also, at present I find it difficult to understand whether the hinge line between A and E was measured and if it wasn’t, why not?

  • Figure 1 is now updated with the 3-leaflet valve scheme. Moreover the proportions and difference between both configurations were shown in this figure. The direct measurement between point A and E was not performed, as it may be calculated from the lengths of respective leaflets and commissures, which were measured.

In Figure 2 to have a photo of a 3-leaflet valve juxtaposed to the current photo.

  • Figure 2 is now updated.

In Figure 3 to have a photo of a 4-leaflet valve juxtaposed to the current photo.

  • Figure 3 is now updated to show both configurations.

Please also consider adding a photo of an ambiguous case; as you know, one person’s commissure (=2 leaflets) is another person’s cleft (=1 leaflet).

  • There are no such ambiguous cases in our study material. The two structures (cleft vs commissure) can be easily distinguished. The term cleft may be used only for congenital, pathologic leaflet splits that may be located in any leaflet and (1) reach the valve annulus or (2) are not connected with the tendinous cords to the papillary muscle. The commissure have also very characteristic appearance, with the presence of the commissural tendinous cord, which originate solely from the respective papillary muscle located beneath the commissure and runs as a single stem that branches radially resembling the ribs of a fan, and thus referred to as a ‘‘fan-shaped chordae.’’

Please consider using a figure like Figure 1 (both 3-leaflet and 4-leaflet schematic), to report your findings (analogous to using a ‘bulls eye plot’ to report on LV 17-segments). Although your text in Results is concise, there is still a substantial number of values and anatomical locations to keep track of and it may be substantially easier if reported in schemes (bold or red font for significantly different values?). Please consider reporting median values (is that not more useful than mean?).

  • Thank you for this comment. We have discussed the possibility of creation of such scheme within all authors and with external experts, unfortunately we have concluded that creation of such ‘bulls eye plot’ is impossible due to of the type and complexity of the data. Instead, the table 1 presents major morphometric observations with clear comparison between both types. Moreover we have updated the Figure 1 – newly created schemes have preserved the relative proportions between structures of the valve.

Do I understand correctly that you have heart weights for each heart but in the linear measurements (Table 1) you have not corrected for heart size? Why not? Are the 4-leaflet hearts larger and may that explain the great dimensions? You do correlate valve perimeter to BMI split on the 3- and 4-leaflet configuration, but why not first do the analysis on all hearts against all BMI, and heart weight? Also, how do your measurements of RAV perimeter (if corrected for body mass) compare to those reported previously (Rowlatt, U. F., Rimoldi, H. J., & Lev, M. (1963). The quantitative anatomy of the normal child's heart. Pediatric Clinics of North America, 10(2), 499-588.).

  • The heart weight was measured for all hearts. No difference in heart weight was observed between two configurations (407.9 ± 86.5 vs. 419.3 ± 75.8 g, P = 0.363). Also no correlations with the heart weight were present. This is now in stated in the results section. Therefore we deem it unreasonable to correct the measurements to the heart weight. The above mentioned publication studied pediatric population (up to 15 years of age), therefore comparisons between Rowlatt et al. study and the current study is doomed to failure.

Even if you find no significant effect or correlation, please still report the output of the statistical tests, e.g. for correlation r and p. For example concerning 4-leaflet configuration (L138), age and sex (L164-165) – as you have done elsewhere (L179).

  • R and P are now reported in line 138. For the L164-165 and L215-217 where all measured parameters of the RAV were compared with age, sex and anthropometric features reporting the exact values for all negative r and p values are unreasonable and simply impossible within the text, therefore we have decided to include the following phrase: (all r < 0.2 and P > 0.05).

Are there in vivo data (echocardiography, MRI, CT) that support your morphometric findings?

  • Unfortunately as a non-clinical department we have not access to clinical imaging facilities such as echocardiography, MRI, CT. However, in our previous work performed on the “functioning” (beating) heart the right atrioventricular valves were analyzed using endoscopic cameras and we were able to describe not only the anatomy, but also a function of the valve. Based on our previous study we are sure that all leaflets (both in 3- and 4-leaflets configuration) are fully functional leaflets.

L47-48: “more” – compared to what, the mitral valve?

  • compared to mitral valve, this is now corrected

L111: okay for linear measurements, but how did you measure curves (hinge lines)?

  • Hinge lines were straightened to form straight line. This is the standard way to measure curves in gross anatomy.

Please note that references 10 and 16 are the same. Please remove one and update the references.

  • Ref. 16 is now updated.

Round 2

Reviewer 1 Report

The authors answered correctly to the questions

Reviewer 2 Report

The revised figures are great.

I now understand a source for confusion to me: that Figure 1 shows the intact anatomy AND measurements, whereas measurements where not done on intact av junctions as in Figure 1 but on opened hearts as in Figure 3. Figure 1 is fine as is, I am not suggesting to change it (as you state, it is standard procedure). But for your next publication you could consider whether a figure of Materials and Methods should not primarily visualize the method, i.e. show Figure 1 with the anatomy as in Figure 3, rather than intact anatomy?

With the number of measurements you have done, there is a very high number of potential comparisons and corrections you can do and I accept that you have chosen a subset of these. That is, it is not imperative to me that you make further changes, but it is not obviously the best choice to compare heart weights of the 3L and 4L groups and then because they are not different to conclude that heart weight cannot explain differences in perimeter length. It would make more sense, it seems to me, to correct each perimeter for heart size and then compare perimeters between groups (and that would also allow you to make use of the Rowlatt analyses if I am not mistaken).      

Best wishes

X